# Simple and Reliable Method for Gastric Subepithelial Tumor Localization Using Endoscopic Tattooing before Totally Laparoscopic Resection

**DOI:** 10.3390/jpm11090855

**Published:** 2021-08-28

**Authors:** Sheng-Fu Wang, Hao-Tsai Cheng, Jun-Te Hsu, Chi-Huan Wu, Chun-Wei Chen, Chun-Jung Lin, Kai-Feng Sung

**Affiliations:** 1Department of Gastroenterology and Hepatology, Chang-Gung Memorial Hospital, Linkou Medical Center, Taoyuan 333, Taiwan; Shanelily@msn.com (S.-F.W.); b9002076@adm.cgmh.org.tw (C.-H.W.); 8902088@adm.cgmh.org.tw (C.-W.C.); ma1249@adm.cgmh.org.tw (C.-J.L.); 2School of Medicine, College of Medicine, Chang-Gung University, Taoyuan 333, Taiwan; hautai@adm.cgmh.org.tw (H.-T.C.); hsujt2813@adm.cgmh.org.tw (J.-T.H.); 3Department of Gastroenterology and Hepatology, New Taipei Municipal TuCheng Hospital, New Taipei City 236, Taiwan; 4Department of General Surgery, Chang-Gung Memorial Hospital, Linkou Medical Center, Taoyuan 333, Taiwan

**Keywords:** gastric subepithelial tumor, tattoo, laparoscopic surgery

## Abstract

Background: Totally laparoscopic surgery for early gastric cancer and subepithelial tumors has been popularized worldwide, yet localization of early or small-sized tumors is a persistent challenge due to difficulty being identified with the lack of manual tactile sensation. Thus, accurate localization with tattooing before the surgery would help improve efficiency during surgery. There are multiple methods to localize tumors before laparoscopy, each with varying advantages and disadvantages. The use of endoscopic tattooing with dye has been carried out for several decades due to its safety, lower cost, and convenience. However, there is a lack of studies on endoscopic tattooing before totally laparoscopic resection. Aims: To evaluate the effect of endoscopic tattooing with dye for gastric subepithelial tumors localization before laparoscopic resection and to evaluate the tattooing effect on different locations of tumors in stomach. Method: We retrospectively collected data of patients with gastric subepithelial tumors who underwent endoscopic tattooing before totally laparoscopic resection from 2017 to 2020 in a university affiliated medical center. All patients were analyzed for preoperative characteristics and then categorized into two groups based on tumor locations concerning the difficulty of laparoscopic surgery. The independent t test and Chi-square test were performed to compare perioperative outcome and complications between these two groups. Result: A total of 19 patients were included retrospectively at our center. The individuals were 5 male and 14 female patients with a mean age of 58.2 years old. Most patients had no symptoms, and the tumors were found incidentally in 12 patients (63%). All tumors were identified clearly during laparoscopic resection. The mean tumor size was 2.3 cm. The surgeries took an average of 111 min and a mean of 7 mL blood loss was found. All tumors had negative resection margins with no recurrence during follow-up. Gastrointestinal stromal tumor was the major pathologic diagnosis, found in 12 patients (63%), followed by the leiomyoma in 5 patients (26%). Only three patients had mild adverse effects after surgery and the symptoms were self-limited. Our analysis found no significant difference in preoperative patient characteristics and perioperative outcomes between patients with differing tumor locations. Conclusion: This study is the first and largest report on endoscopic tattooing with dye before laparoscopic resection of gastric subepithelial tumor resection. Our results emphasize that endoscopic tattooing with dye is a safe and reliable method for localizing subepithelial tumors in the stomach prior to totally laparoscopic resection, with no correlation to where the tumor is located.

## 1. Introduction

Early gastric cancer or gastric subepithelial tumors in selective patients can be resected safely by using endoscopic mucosal resection (EMR) or endoscopic submucosal dissection (ESD) [1,2]. The current indications of ESD for early gastric cancer, suggested by Japanese Gastric Cancer Association (JGCA), includes (I) Intramucosal tumor; (II) well-differentiated intestinal tumor type according to Lauren; (III) tumor size < 2 cm; (IV) absence of neoplastic ulcer; (V) absence of lymphovascular invasion; and (VI) negative horizontal and deep margins [3]. In addition, the gastric subepithelial tumor is suggested to be followed up regularly if the tumor size is less than 2 cm due to low malignant potential, whilst resection is indicated if the tumor is more than 3 cm and endoscopic ultrasound showed nodular change, heterogeneous pattern, or anechoic area [4]. Moreover, some patients who worry about potential malignant change or are tired of annual endoscopic follow up, may also request resection. Although ESD for subepithelial tumors has an acceptable, complete resection rate, there are risks involved, including bleeding, tumor spillage, or perforation, depending on which layer the tumor had originated [5]. Thus, surgery would be an alternative method especially if the subepithelial tumor is located at the muscularis propria.

In recent years, minimally invasive surgery by totally laparoscopic resection has improved postoperative recovery of patients who suffered from gastric subepithelial tumors. Moreover, laparoscopic partial gastrectomy or wedge resection for gastric submucosal neoplasms increases patients’ quality of life post gastrectomy as it preserves the residual stomach. However, there are some limitations of totally laparoscopic surgery. Tumor size is a matter. Large tumors usually require more manipulation, increasing the risk of rupture, and thus laparoscopic surgery is only indicated for small subepithelial tumors (diameter ≤ 2 cm) [6]. Conversely, Otani et al. and Ryu et al. noted that laparoscopic resection of subepithelial tumors could be feasible with tumors up to 5 cm [7,8]. In addition, tumor site may impact the difficulty and duration of surgery. Surgery may be more challenging and time consuming if the tumor is located at the posterior wall or esophagogastric junction (EGJ) [9,10].

Therefore, an accurate localization would be essential to facilitate the operation. Usually, small subepithelial tumors with an intraluminal growth are difficult to be localized by tactile sensation during laparoscopic surgery, thus preoperative localization is required.

There are nine methods for tumor localization during laparoscopic gastric surgery [11]. Three methods are used as present practice. Endoscopic tattooing is the first method and is very convenient as it can be done within 3 days prior to laparoscopic surgery [12]. The second method is endoscopic marking clip, which was applied for localization in early gastric cancer in previous studies [13,14]. However, the clips can only be visualized by fluoroscopy and may prolong operative time in the operative room. The third method used by Hideo Matsui et al., endoscopy-assisted gastric resection during laparoscopic surgery in gastric cancer, is also a reliable procedure [15]. However, an additional workforce of endoscopists and technicians is required and is needed to standby at the operative room.

Tattooing with dye before surgery by endoscopy was the first method developed with its safety and lower cost and is still commonly used for colonic lesions and early gastric cancer at present [12,16]. However, there is a lack of studies on tattooing with dye in gastric subepithelial tumors before totally laparoscopic surgery. In this study, we will report our experience about endoscopic tattooing with dye for gastric subepithelial tumors prior to totally laparoscopic surgery.

## 2. Materials and Methods

### 2.1. Patients

This retrospective cohort study was conducted from January 2017 to June 2021 at Chung Gang Memorial hospital in Taiwan. Patients’ data were retrieved from our prospectively registered database in the therapeutic endoscopic center. Patients with endoscopic diagnosis of gastric subepithelial tumors and tattooing before laparoscopic gastrectomy were enrolled. Data on consecutive patients were extracted from the database that included patient characteristics (age, gender), preoperative data (tumor size, location, tattooing time to surgery), surgical outcomes including intraoperative events (surgical approach, operative time, and estimated blood loss), pathological features (pathological diagnosis, specimen size/ratio, mitosis rate), and postoperative course and outcome. Patients who received other methods such as combined endoscopic metal clips localization or endoscopic tattooing simultaneously at operative room with laparoscopy were excluded. All patients’ data were collected to evaluate the effect of endoscopic tattooing with dye for tumor localization before laparoscopic gastric resection. The patients were then divided according to the positions of the tumors in the stomach (anterior wall, greater curvature, fundus and posterior wall, cardia, lesser curvature) in accordance to the difficulty of laparoscopic approach, followed by an evaluation of the effects of tattooing on different locations of tumors in stomach.

### 2.2. Localization Method

The endoscopic tattooing was done during upper gastrointestinal endoscopy at our therapeutic endoscopic center the same day or 1 day before surgery. The patients briefly laid in left lateral posture under conscious sedation with fentanyl and midazolam. The scope then localized the margin of the gastric subepithelial tumors. Tattooing was performed with a carbon particle containing solution, SPOT (GI Supply, Camp Hill, PA, USA) without dilution. The 23 Gauge injection needle (Olympus, product number: NM-400L-0423, Tokyo, Japan) was punctured as perpendicularly as possible at four quadrants of the tumors deep into the muscle layer with 0.1 mL SPOT injected in each quadrant (Figure 1). Patients then went into laparoscopic surgery as scheduled. The laparoscopic gastric surgery was done by surgeons at the operating room. Surgical procedures, laparoscopic wedge resection, or laparoscopic subtotal gastrectomy was performed as the surgeon planned.

### 2.3. Surgery

All the procedures were done through totally laparoscopy wedge resection or partial gastrectomy. A four-port technique was briefly used. The initial port was placed through a 2.5-cm infraumbilical incision made using the open method and a commercially available access port (EZ Access; Hakko, Nagano, Japan). During the procedure, a pneumoperitoneum was established using carbon dioxide insufflation at a pressure of approximately 10–12 mmHg according to the body type of the patient. For tumors located away from the pylorus or antrum, we performed a wedge resection using a linear stapler (Ethicon Endo-Surgery, Cincinnati, OH, USA) after dividing the surrounding vessels around the marked area, including tumor with an energy device (LigaSure^TM^). For the lesion located near the pyloric area, distal gastrectomy (antrectomy) followed by gastrojejunostomy was carried out without omentectomy and lymphadenectomy. The staple line or gastrojejunostomy was reinforced with absorbable suture. One Jackson Pratt drain was placed around the resection site. During laparoscopic gastrectomy, the blue dye was inspected directly from the serosa (Figure 2A). The tumor was then resected along the margin of the tattoo with a stapler smoothly (Figure 2B).

This study is approved by institution review board from Chang Gung Memorial Hospital (000).

### 2.4. Statistical Analysis

The univariate analysis was done using Chi-square test for categorical variables and the independent sample *t* test for continuous variables. A *p* value less than 0.05 was considered significant. Variables are expressed as mean plus range. All statistical analyses were performed using the Statistical Product and Service Solutions, SPSS, version 26 (IBM, Armonk, NY, USA).

## 3. Result

We collected data from 21 patients with gastric subepithelial tumors who underwent endoscopic tattooing with SPOT and totally laparoscopic surgery between 2017 and 2021. We excluded one patient who simultaneously underwent combined endoscopic tattooing with laparoscopic surgery and another patient who refused operation after endoscopic tattooing, resulting in 19 patients enrolled in this study. The basic characteristics of patients are presented in Table 1 and Table 2. The mean age of these patients was 58.2 years old and was predominately female (M:F = 5:14). Most patients had no symptoms and the tumors were found incidentally in 12 patients (63%). Tumor sizes were all less than 3 cm by endoscopy, endoscopic ultrasound (EUS), or computer tomography (CT) except patient No. 7. All of these subepithelial tumors were found to have arisen from the muscularis propria under EUS. Sixteen patients received tattooing procedures on the same day of surgery and three cases were performed 1 day before surgery. All tattoo dye was identified clearly during laparoscopic resection. Eighteen patients received wedge resection and only one patient received partial gastrectomy. The mean operative time is 111 min with an estimated average of 7 mL blood loss. All tumors had a negative resection margin and no recurrence during follow up. Based on the histopathological study, the mean tumor size was 2.3 cm. Gastrointestinal stromal tumor (GIST) was the major pathologic diagnosis found in 12 patients (63%), followed by the leiomyoma in 5 patients (26%). Only three patients had mild adverse effects after surgery and the symptoms were self-limited. Two cases had mild fever without sepsis or abdominal pain, which likely occurred from post-intubation fever. One case had coffee ground material about 20~30 mL from the nasogastric tube drainage with self-limited symptoms but may not be related to the tattooing. None of the cases had perforation nor required second surgery.

Furthermore, we categorized these patients into two groups depending on the tumor location. Group 1 included tumors located at the anterior wall, greater curvature side or fundus; and group 2 included the posterior wall, lesser curvature side, near pylorus, or EGJ. A tumor on the anterior gastric wall or greater curvature side is easy to visualize and remove. However, it is both difficult and time-consuming to excise a gastric tumor from the posterior wall or near the pylorus and esophagogastric junction [17]. The comparison of surgical outcomes is presented as Table 3. The peri-operative outcomes also had no significant differences with operative time, blood loss, post-operation complication, time to intake, and hospital days. Regarding the specimens, all these cases had negative margin resections and there is no significant difference between the two groups with their distance from the margin. There is no clinical significance between the maximal diameter of the tumor, maximal diameter of resected stomach, or the ratio of the tumor/resected stomach. We also classified these patients into two groups based on tattooing on the same day of surgery and on the day before surgery with the surgical outcome shown as Table 4. There is no clinical significance between the two groups. No recurrence was found under endoscopy surveillance or CT image during following up (mean 10.6 months, 2–24 months).

## 4. Discussion

Based on our results, most patients were detected incidentally, and diagnosed as GIST or leiomyoma under EUS depending on the echogenicity and the layer the tumor had arisen from (all from muscularis propria). Moreover, the sizes of most subepithelial tumors were less than 2 cm and there were no worrisome features under EUS. In other words, these patients were supposed to be followed up every 3~12 months if asymptomatic or have diagnostic intervention including jumbo biopsy, bite on bite, submucosal core biopsy or EUS-guided fine needle aspiration cytology (EUS-FNA) or biopsy (EUS-FNB) according to ASGE guidelines [18]. Several studies suggested EUS-FNB should be the first line modality when comparing EUS-FNA. A recent large multicenter study that enrolled 229 patients concluded that EUS-FNB was superior to EUS-FNA in the sensitivity and accuracy of diagnosis in subepithelial tumors [19]. Another two studies also revealed that EUS-FNB outperformed EUS-FNA in the evaluation of small pancreatic neuroendocrine tumors or solid pancreatic lesions even without rapid on-site evaluation [20,21]. However, studies on EUS-FNB of subepithelial tumors measuring less than 2 cm have yielded inconsistent results [22,23]. Regarding the utility of EUS-FNB for tumors less than 2 cm, Hedenström et al. [24] reported as beneficial, while Fujita et al. [25] reported as contrary. Thus, Mitsuhiro Kida et al. suggested patients could be followed up until tumors grow larger than 1~2 cm before EUS FNA/FNB should be conducted [4]. However, patients with small-sized subepithelial tumors may be concerned about malignant potential and weary of repeated endoscopy, and therefore surgical resection would be an option.

During totally laparoscopic surgery, tumors are hardly identified due to their small size, and cannot be palpated by hand, thus localization before surgical resection is essential.

There are nine methods for tumor localization during laparoscopic gastric surgery [11]. Endoscopic dye tattooing method is the first and is convenient. It could be done within 3 days prior to laparoscopic surgery [12]. It omits the necessity of expensive instruments and extra workforce in the operative room and, therefore, shortens the operative time. Endoscopic autologous blood tattooing has the same benefit of dye tattooing but the blood may spread less due to its viscosity. Pre-operative three-dimensional computed tomography (3D-CT) reconstruction has the disadvantages of requiring endoscopic clipping before CT scan and cooperation with a radiologist for 3D reconstruction [11,26]. Endoscopic clipping needs intraoperative X-ray or laparoscopic ultrasonography for localization due to the lack of tactile sensation inherited from total laparoscopic surgery [27]. Three recently developed methods, the magnetic clip, fluorescent clip, and radio-frequency identification clip, need expensive detection systems in the operative room [14,28,29]. Intra-operative endoscopy itself is simple, however, endoscopic experts and technicians are required.

SPOT (GI Supply, Camp Hill, PA, USA), similar to India ink [30], is a sterile and biocompatible suspension containing highly purified, very fine carbon particles and was developed specifically for endoscopic tattooing [31]. It is also a U.S. Food and Drug Administration (FDA)-certified product for tattooing. So we utilize the advantages of SPOT, such as safety, less inflammation than India ink, and without the need for dilution to localize subepithelial tumors [11]. Furthermore, there are no studies that report on SPOT in gastric subepithelial tumor localization, and information is scarce, thus we hope to verify this reliable and convenient technique with our experience.

In our analysis, all patients had negative resection margins with no recurrence, which implies that this method could be reliable in gastric subepithelial tumor localization. The common drawback of this dye is its substantial peritoneal staining when unintended transmural injection occurs. The peritoneal staining can obscure surgical dissection planes, making surgery more dangerous and challenging [32,33]. According to one previous study, 63 patients underwent colonic tattooing with SPOT and six patients had localized leakage found during operation. Five of them were symptomless and only one case had chillness without abdominal pain or fever [34]. However, only 0.22% post-tattooing complications with India ink was reported in another large study that included 447 patients [32]. McArthur et al. [35] also reported that complications in a study of 195 patients who underwent endoscopic tattooing with India ink for colonic lesions were rare. These studies suggest endoscopic tattooing with dye is a safe procedure in colon lesions, not to mention that SPOT is a more harmless product and the gastric wall is thicker than the colon. In our study, there was no peritoneal dye leakage, which may be due to different thicknesses of the stomach and colon wall. There were only three cases that suffered from mild adverse effects, which re likely unrelated to the tattooing itself, as no localized leakage was found during surgery. Most patients started to eat within 3 days and were discharged in 1 week, implying that this tattooing method is safe for gastric subepithelial tumor localization.

Another crucial point is the depth of injection in during endoscopic tattooing. An ideal technique should avoid possible complications from transmural injection, which results in spillage and peritoneal inflammation. In addition, the tattooing around the subepithelial tumors may be invisible if the injection was only done on the submucosa. To prevent these drawbacks, Hyman et al. recommended a technique of “four quadrant” circumferential tattooing to advance intraoperative visualization of colon lesions [36]. The technique requires the injection of 0.2–0.5 mL of dye around the lesion, and the insertion of the needle tangentially to prevent transmural injection. Our study differs as it concerns the stomach, in which the gastric wall is thicker than that of the colon.

If the needle is inserted tangentially, only submucosal injection would occur. Surgeons cannot inspect the tattoo from the serosa. Based on our study, we recommend the needle to be inserted perpendicularly with just 0.1 mL of SPOT and do not push forward hard, as it is enough for visualization without spreading out. Widespread stain with more SPOT volume would lead to unnecessary resection of the stomach.

The location of tumors may increase the difficulty of laparoscopic surgery. According to previous studies, gastric GIST, which is located at the posterior wall and near the gastroesophageal junction, is much more challenging to resect than the anterior wall [37]. Thus, we classified our patients into two groups based on the location of tumors to recognize whether tattooing can facilitate the procedure even if the tumor is located at the posterior wall. In our study, no clinical significance was found in the duration of surgery, blood loss, size of the tumor or resected stomach, tumor/resected stomach ratio, and other perioperative factors between these two groups. Namely, tattooing with SPOT makes no difference in totally laparoscopic surgery regardless of the locations of subepithelial tumors.

We also found no difference in surgical outcome whether the tattoo was done on the day of surgery or 1 day before surgery. This may be attributable to SPOT’s inability to be absorbed by the serosa, thus the dye would be retained till operation. A previous study reported that the tattooing using SPOT for colonic lesion localization was still identifiable after 3~12 months [38]. However, the dye injected into the subserosal layer may have had lateral, widespread diffusion, which then caused the margin to expand. In our data, the resected margin from the tumor was more distant in the group that underwent tattooing 1 day before surgery, which likely resulted from widespread diffusion although the statistics showed no clinical significance (*p* = 0.078). Therefore, tattooing on the day of surgery is recommended in our experience.

In conclusion, this is the first report in which we show that tattooing with SPOT for subepithelial tumors before totally laparoscopic surgery may be a safe and reliable method regardless of the tumor location with the tattooing procedure done on the same day or 1 day before surgery with limited evidence.

There are some limitations in our study. First, only 19 patients are enrolled in our study within 3.5 years. To our best knowledge, this is the largest case series report on endoscopic tattooing with dye before laparoscopic gastrectomy for gastric subepithelial tumors. Considering that most gastric subepithelial tumors are benign and follow-up is recommended, small tumors rarely underwent resection. To validate our experience, a cooperative study between multiple centers is expected in the future. Second, this is a single center retrospective study with a limited patient number. Thus, we could not categorize patients into every position where the tumor was located at in the stomach. Third, comparisons with other methods of tumor localization were not performed as we mostly use endoscopic tattooing with SPOT with patients at our hospital. 

## Figures and Tables

**Figure 1 jpm-11-00855-f001:**
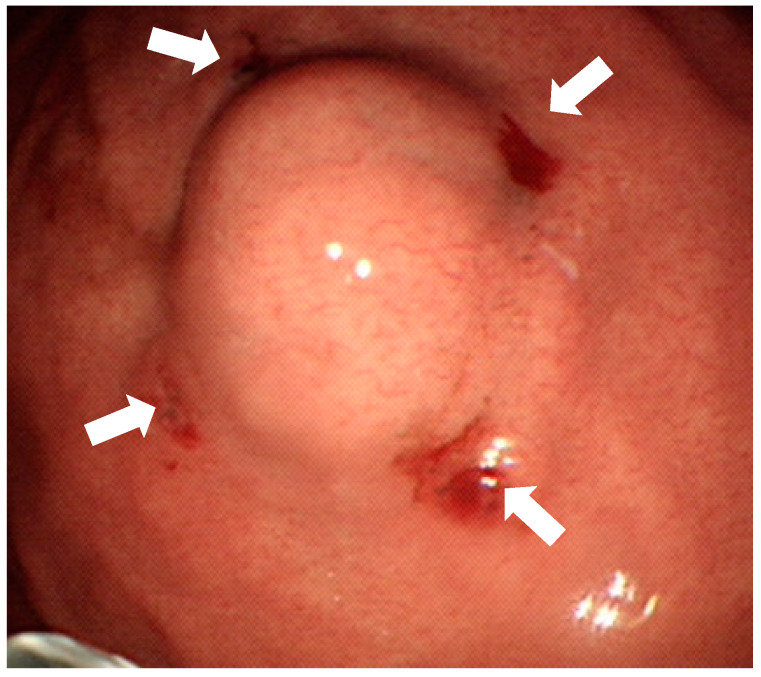
Endoscopic tattooing with SPOT before laparoscopic resection of the subepithelial tumor. The “four quadrants” method is used so that 0.1 mL SPOT is injected into muscularis propria layer by perpendicular injection (white arrows). Only puncture holes are left and there is no submucosal injection.

**Figure 2 jpm-11-00855-f002:**
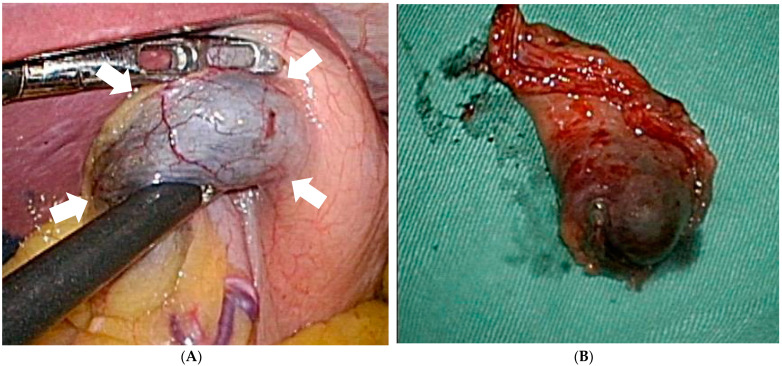
The tattoo is inspected directly from serosa (white arrows) during laparoscopic surgery without spillage (**A**) and then the tumor is resected completely (**B**).

**Table 1 jpm-11-00855-t001:** Patient detail.

Pt No	M/F	Age	Initial Symptom	Tumor Location	Time to Op (Hour)	EUS Layer	Endoscopy Size (cm)	Pathological Diagnosis	Stomach Diameter (cm)	Tumor Diameter (cm)	Diameter Ratio	Surgery Time (min)	Complication	Hospital Days	Time to Intake (Day)
1	F	70	No symptom	Fundus	20.5	4	1.2	Leiomyoma	4.0	2.2	0.550	90	No	5	2
2	F	51	Fullness	Fundus	3.5	not done	3	GIST	6.0	3.5	0.583	111	Fever	5	2
3	F	43	No symptom	Fundus	3.5	4	1.2	GIST	2.9	1.6	0.552	120	No	3	1
4	F	44	Epigastric pain	Posterior wall	5	4	1.5	Leiomyoma	6.6	1.8	0.273	105	No	8	4
5	M	70	No symptom	Posterior wall	8	not done	2	GIST	6.5	5.7	0.877	120	No	11	3
6	F	57	Epigastric pain	Anterior wall	3.5	4	2	GIST	5.1	2.5	0.490	90	No	6	4
7	F	63	No symptom	Lesser curvature	7	4	2	GIST	4.5	3.7	0.822	110	No	6	5
8	F	44	No symptom	Posterior wall	4.5	4	1.2	GIST	3.1	1.4	0.452	120	No	4	2
9	F	66	No symptom	Greater curvature	2.5	4	2	Glomus tumor	3.0	2.0	0.667	120	No	8	3
10	M	70	UGI bleeding	Posterior wall	4	not done	3	GIST	5.3	2.5	0.472	150	Fever	10	4
11	F	64	UGI bleeding	Fundus	23	4	3	GIST	7.8	3.2	0.410	140	No	6	2
12	F	50	No symptom	Posterior wall	3	4	1	Leiomyoma	6.4	0.9	0.141	70	No	3	2
13	M	50	No symptom	Posterior wall	3	4	2	Ectopic pancreas	2.8	1.4	0.500	100	No	5	4
14	F	70	No symptom	Greater curvature	3	4	1.5	GIST	2.8	1.6	0.571	110	No	6	3
15	M	52	Epigastric pain	Anterior wall	6.5	4	1	Leiomyoma	4.1	0.9	0.220	60	No	9	5
16	M	50	No symptom	Fundus	3	4	1.4	GIST	5.3	1.2	0.226	130	No	6	3
17	F	62	No symptom	Cardia	2.5	4	3	Leiomyoma	4.5	2.4	0.533	150	No	7	3
18	F	57	No symptom	Lesser curvature	22	not done	2	GIST	3.6	2.1	0.583	105	No	5	3
19	F	73	No symptom	Fundus	2	4	2	GIST	3.6	2.5	0.694	100	No	5	3

GIST: Gastrointestinal stromal tumor.

**Table 2 jpm-11-00855-t002:** Patient characteristics.

	Total	AW + GC + Fundus (*n* = 10)	PW + LC + Cardia (*n* = 9)	*p* Value
Age(year)	58.2 ± 10.1	59.6 ± 10.3	56.7 ± 10.2	0.456
Gender				0.434
Male	5 (26.3%)	2 (20.0%)	3 (33.3%)	
Female	14 (73.7%)	8 (80.0%)	6 (66.7%)	
Tumor size(cm)				
Endoscopy	1.9 ± 0.7	1.8 ± 0.7	2.0 ± 0.7	0.634
EUS	1.8 ± 0.5	1.8 ± 0.6	1.6 ± 0.4	0.976
CT	2.2 ± 1.3	2.2 ± 1.2	2.2 ± 1.5	0.749
Specimen	2.3 ± 1.2	2.1 ± 0.8	2.4 ± 1.5	0.921
Pathology diagnosis				0.478
leiomyoma	5 (26.3%)	2 (20.0%)	3 (33.3%)	
GIST	12 (63.2%)	7 (70.0%)	5 (55.6%)	
Glomus tumor	1 (5.3%)	1 (10.0%)	0 (0.0%)	
Ectopic pancreas	1 (5.3%)	0 (0.0%)	1 (11.1%)	
Mitosis of GIST (/50HPF)	1.5 ± 1.8	1.8 ± 2.1	1.2 ± 1.4	0.509
Time to surgery (hour)	6.8 ± 6.8	7.1 ± 7.8	6.6 ± 6.1	0.508
Procedure of surgery				0.950
Partial gastrectomy	1	1 (10.0%)	0 (0%)	
Wedge resection	18	9 (90.0%)	9 (90%)	

AW: anterior wall; PW: posterior wall; GC: greater curvature; LC: lesser curvature; EUS: endoscopic ultrasound; CT: computed tomography; GIST: gastrointestinal stromal tumor.

**Table 3 jpm-11-00855-t003:** Perioperative outcome.

	AW + GC + Fundus (*n* = 10)	PW + LC + Cardia (*n* = 9)	*p* Value
Margin free (%)	10 (100%)	9 (100%)	1.000
Distance from margin (cm)	0.5 ± 0.3	0.5 ± 0.4	0.795
Specimen size			
Maximal diameter of stomach (cm)	4.5 ± 1.6	4.8 ± 1.5	0.509
Maximal diameter of tumor (cm)	2.1 ± 0.8	2.4 ± 1.5	0.921
Ratio of diameter (tumor/stomach)	0.5 ± 0.1	0.5 ± 0.2	0.922
Duration of operation (min)	107.0 ± 23.0	114.0 ± 25.0	0.644
Blood loss (mL)	5.2 ± 1.9	10.6 ± 14.9	0.296
Complication			0.950
No	8 (80.0%)	8 (88.8%)	
Yes			
Leakage	0 (0.0%)	0 (0.0%)	
Fever	1 (10.0%)	1 (11.1%)	
Hospital days (day)	5.9 ± 1.7	6.6 ± 2.7	0.763
Time to intake (day)	2.8 ± 1.1	3.3 ± 1	0.324
Follow up (month)	10.1 ± 6.8	11.2 ± 6.2	0.589
Time to last image (month)	12.6 ± 8	9.8 ± 6.2	0.648
Recurrence			1.000
Yes	0 (0.0%)	0 (0.0%)	
No	10 (100.0%)	9 (100.0%)	

**Table 4 jpm-11-00855-t004:** Surgical outcome.

	Tattoo on Day of Surgery (n = 16)	Tattoo on Day Before Surgery (n = 3)	*p* Value
Distance from margin (cm)	0.4 ± 0.3	0.8 ± 0.1	0.078
Specimen size			
Maximal diameter of stomach (cm)	4.5 ± 1.4	5.1 ± 2.3	0.542
Maximal diameter of tumor (cm)	2.2 ± 1.2	2.5 ± 0.6	0.716
Ratio of diameter (tumor/stomach)	0.5 ± 0.2	0.5 ± 0.1	0.938
Duration of operation (min)	110.3 ± 24.1	111.6 ± 25.6	0.943
Blood loss (mL)	8.4 ± 11.2	4.0 ± 1.7	0.513

## Data Availability

Not applicable.

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
