# Peer review of "Simple and Reliable Method for Gastric Subepithelial Tumor Localization Using Endoscopic Tattooing before Totally Laparoscopic Resection"

_jpm, 2021, doi:10.3390/jpm11090855_

Round 1

Reviewer 1 Report

In this study, the Authors reported a large case series of patients underwent endoscopic tattooing of subepithelial gastric lesions, in order to facilitate laparoscopic resection.

Here my comments:

In 6 cases, surgical resection was not indicated because of benign lesions (5 leiomyomas and 1 ectopic pancreas). Why EUS-FNB was not performed in these cases? Moreover, in the discussion section, a deeper comment on the yield of EUS-FNB in comparison with FNA in these type of lesions should be included. Doing so, cite PMID 34116031 and PMID 33390343.

Please, spell abbreviations in the tables.

English language should be extensively improved.

Author Response

Response to Reviewer 1 Comments

Point 1:

In 6 cases, surgical resection was not indicated because of benign lesions (5 leiomyomas and 1 ectopic pancreas). Why EUS-FNB was not performed in these cases? Moreover, in the discussion section, a deeper comment on the yield of EUS-FNB in comparison with FNA in these type of lesions should be included. Doing so, cite PMID 34116031 and PMID 33390343.

Response 1:

Thank you very much for the critical comments and suggestions.  In these 6 cases, the initial diagnosis based on endoscopic ultrasound were leiomyoma or gastrointestinal tumor because of hypoechoic pattern and all these tumors were arisen from 4th layer. After the surgery, pathology then confirm the benign lesions of these 6 patients. In fact, all of patients in our study could be followed up every 3~12 months or doing EUS-FNA/FNB depending on ASGE guideline[1], but these patients choose to have surgical resection after discussing with their doctors because weary of endoscopic followed up and the possibility of false negative with biopsy.  Another reason is economic issue that the patients had to pay the fee of EUS-FNA/FNB about one thousand USD in Taiwan while insurance could cover the whole fee of the surgery.

We’ve added the comment on the yield of EUS-FNB in comparison with FNA and we also cited the PMID 34116031 and PMID 33390343 in the text. (Page 16, Line 9-15)

Point 2:

Please, spell abbreviations in the tables.

Response 2:

Thank you very much for the critical comments and suggestions. In order to keep the table concise, we’ve added the full name of abbreviations below the tables.

Reference:

  1. Standards of Practice, C., et al., The role of endoscopy in subepithelial lesions of the GI tract. Gastrointest Endosc, 2017. 85(6): p. 1117-1132.

Reviewer 2 Report

Comments to Author:

The authors reported the usefulness of endoscopic tattooing for surgery of gastric submucosal tumors. While this is an interesting retrospective study, there are several concerns to be resolved.

Major concerns:

  1. As the author mentioned, there is too small sample size to establish the conclusion.
  2. Did you perform endoscopy during the surgery? If so, what is the advantage of the tattooing?
  3. Does the tattooing method need four quadrants? If identifying the tumor location is merit, only two or three spots are enough to do that.
  4. In Figure 2, SPOT dye was spreading only under the tumor. Why didn’t the dye spread outside of the tumor?
  5. In page 10, one case had coffee ground material from the nasogastric drainage, did that mean bleeding or SPOT dye? How much mount was that? I feel that was not complication.
  6. The pathological tumor size should be described in Table 2.
  7. The authors recommended this tattooing is performed on the day of surgery, there is few advantages comparing with performing endoscopy during the surgery. How do you explain that?
  8. All cases divided into two groups, however, there are no significant differences in all points. How about dividing the tumor location into upper and lower stomach?

Minor concerns:

There are several typographical errors in the manuscript and Table.
Page 10, line 217, nasogastric; “n” should be small letter.
Table 1, you should explain the meaning of “nil” and abbreviation of “NG, nasogastric tube”.

Author Response

Response to Reviewer 2 Comments

Major comments:

Point 1:

As the author mentioned, there is too small sample size to establish the conclusion.

Response 1:

Thank you very much for the critical comments and suggestions. We’ve modified the conclusion in the text (Page: 19 Line:16,18)

Point 2:

Did you perform endoscopy during the surgery? If so, what is the advantage of the tattooing?

Response 2:

Thank you very much for the critical comments and suggestions. We do performed endoscopy to localize tumor during surgery previously, but needs of an extra manpower with a doctor and assistant to execute this procedure. Tattooing before the laparoscopic surgery was done at the scheduled time in our endoscopic center without needs of additional time and membership.

Point 3:

Does the tattooing method need four quadrants? If identifying the tumor location is merit, only two or three spots are enough to do that

Response 3:

Thank you very much for the critical comments and suggestions. The tattooing of the subepithelial lesion may be invisible due to superficial injections. For preventing these drawback, Hyman et al. recommended a technique of “four quadrant” circumferential tattooing to advance intraoperative visualization[1]. We’ve mentioned it in the text ( Page 18, Line 13-22) and we perform this procedure based on their suggestion. In addition, wedge resection is conducted for small size tumors with preserving stomach as much as possible thus the tattooing margin is very important for complete resection, “four quadrant” method may be much helpful for this kind of surgery. Two or three spots tattooing may be feasible in the future but further validation is needed, thanks for your constructive suggestion and we will take it into consideration in the following essay.

Point 4:

In Figure 2, SPOT dye was spreading only under the tumor. Why didn’t the dye spread outside of the tumor?

Response 4:

            There are two important points of our tattoo injection technique. First, the direction of the injection needle should be perpendicular to the mucosa surface but we don’t push forward hardly in avoid of piercing the stomach. Second, we injected only 0.1 ml SPOT solution without dilution per shot. These techniques limited the spread of the ink in subserosa layer. Another reason is the wall of stomach is more thicker than colon thus previous study revealed cases with dye spreading out when treating colonic lesion[2] but we didn’t encounter this complication. We’ve also modified the text (Page 18 Line 26).

Point 5:

In page 10, one case had coffee ground material from the nasogastric drainage, did that mean bleeding or SPOT dye? How much mount was that? I feel that was not complication

Response 5:

Thank you very much for the critical comments and suggestions. The patient had only 20~30mL coffee ground material in the nasogastric tube without decreased haemoglobin or vital sign change with self-limited symptom. It may be due to anastomosis oozing or stress ulcer and possibly not related to SPOT tattooing, thus we’ve deleted it in the Table 1. and Table 3., we also modified the text (Page 10 Line 24,25).

Point 6:

The pathological tumor size should be described in Table 2.

Response 6:

Thank you very much for the critical comments and suggestions. We’ve added it in the Table 2.

Point 7:

The authors recommended this tattooing is performed on the day of surgery, there is few advantages comparing with performing endoscopy during the surgery. How do you explain that?

Response 7:

Before we develop the tattooing technique for submucosal tumor resection, we used intra-operative endoscopy for tumor localization. However, one endoscopic doctor and one endoscopic technician had to move our equipment to operation room. During the operation, it is the business time of our endoscopic center and it took hours for this intra-operative procedure, including stand-by at the operative room and movement. If the patient was sent to our endoscopic center during the morning of surgery day, it only took 15-20 minutes for the tattooing procedure. This is the main advantage of this method.

Point 8:

All cases divided into two groups, however, there are no significant differences in all points. How about dividing the tumor location into upper and lower stomach?

Response 8:

Thank you very much for the critical comments and suggestions. We’ve analysed the perioperative outcome and surgical outcome based on tumor location dividing to upper and lower part but still showed no clinical significance as below. It may be due to small number patients or the tattooing is really helpful to ameliorate surgical outcome even difficult location. The statistical result is shown as below.

Levene's test for equality of variance

T test for whether the mean values are equal

F

Significance

T

df

Significance (both sides)

Average difference

Standard error

95% confidence interval for the number of differences

Lower limit

Upper limit

time to op (hour)

Use equal variance

1.134

.302

.005

17

.996

.0167

3.2480

-6.8361

6.8694

Do not use equal variance

.005

16.581

.996

.0167

3.2001

-6.7479

6.7813

endoscopy size(cm)

Use equal variance

6.309

.022

1.638

17

.120

.4967

.3032

-.1430

1.1363

Do not use equal variance

1.685

14.606

.113

.4967

.2948

-.1331

1.1265

CT size(cm)

Use equal variance

.000

.988

.577

17

.571

.3722

.6446

-.9878

1.7322

Do not use equal variance

.574

16.224

.574

.3722

.6485

-1.0010

1.7454

mitosis(/50HPF)

Use equal variance

2.800

.113

1.482

17

.157

1.2111

.8173

-.5132

2.9354

Do not use equal variance

1.515

15.644

.150

1.2111

.7994

-.4866

2.9089

stomach diameter(cm)

Use equal variance

.173

.683

.307

17

.762

.2189

.7122

-1.2838

1.7216

Do not use equal variance

.307

16.607

.763

.2189

.7139

-1.2901

1.7279

tumor diameter

Use equal variance

2.602

.125

.200

17

.844

.1089

.5453

-1.0416

1.2594

Do not use equal variance

.193

10.962

.851

.1089

.5656

-1.1364

1.3542

diameter ratio

Use equal variance

5.377

.033

.049

17

.961

.004547

.092000

-.189555

.198649

Do not use equal variance

.048

11.204

.963

.004547

.095286

-.204711

.213805

distance from margin(cm)

Use equal variance

1.626

.219

.141

17

.889

.0222

.1574

-.3099

.3544

Do not use equal variance

.137

12.675

.893

.0222

.1617

-.3281

.3726

op time(min)

Use equal variance

.214

.649

1.750

17

.098

17.9889

10.2811

-3.7024

39.6802

Do not use equal variance

1.754

16.912

.098

17.9889

10.2586

-3.6635

39.6413

blood loss(mL)

Use equal variance

4.439

.050

-1.131

17

.274

-5.3556

4.7355

-15.3467

4.6355

Do not use equal variance

-1.071

8.243

.314

-5.3556

4.9988

-16.8239

6.1127

hospital days

Use equal variance

3.178

.093

-.645

17

.527

-.6556

1.0160

-2.7991

1.4880

Do not use equal variance

-.633

14.012

.537

-.6556

1.0364

-2.8783

1.5672

time to intake(day)

Use equal variance

.554

.467

-1.561

17

.137

-.7444

.4770

-1.7508

.2619

Do not use equal variance

-1.546

15.736

.142

-.7444

.4816

-1.7669

.2780

f/u months

Use equal variance

1.043

.321

-1.038

17

.314

-3.0222

2.9127

-9.1674

3.1230

Do not use equal variance

-1.056

16.320

.306

-3.0222

2.8627

-9.0812

3.0367

Minor comments

Point 1

There are several typographical errors in the manuscript and Table. Page 10, line 217, nasogastric; “n” should be small letter. Table 1, you should explain the meaning of “nil” and abbreviation of “NG, nasogastric tube”

Response 1:

Thank you very much for the critical comments and suggestions. We’ve correct nasogastric “n” to small letter (Page 10 Line 24). We’ve correct “nil” to “not done” in the Table 1. “NG coffee ground” had been deleted in Table 1.

Reference:

  1. Hyman, N. and J.D.J.G.e. Waye, Endoscopic four quadrant tattoo for the identification of colonic lesions at surgery. Gastroenterol Endosc, 1991. 37(1): p. 56-58.
  2. Park, J.W., et al., The usefulness of preoperative colonoscopic tattooing using a saline test injection method with prepackaged sterile India ink for localization in laparoscopic colorectal surgery. Surg Endosc, 2008. 22(2): p. 501-5.